# Development and Evaluation of Curcumin Liquid Crystal Systems for Cervical Cancer

**Sheba R David [1]**, **Nurul Akmar Binti Anwar [2]**, **Koh Rhun Yian [3]**, **Chun-Wai Mai [2]**, **Sanjoy Kumar Das [4]** and **Rajan Rajabalaya [1],***

[1] PAPRSB Institute of Health Sciences, Universiti Brunei Darussalam, Jalan Tungku Link, Bandar Seri Begawan BE1410, Brunei; sheba.david@ubd.edu.bn
[2] School of Pharmacy, International Medical University, No. 126, Jalan Jalil Perkasa 19, Bukit Jalil, Kuala Lumpur 57000, Malaysia; akmar101@yahoo.com (N.A.B.A.); chunwai_mai@imu.edu.my (C.-W.M.)
[3] School of Health Sciences, International Medical University, No. 126, Jalan Jalil Perkasa 19, Bukit Jalil, Kuala Lumpur 57000, Malaysia; rhunyian_koh@imu.edu.my
[4] Institute of Pharmacy, Jalpaiguri, Govt. of West Bengal, West Bengal 735101, India; sanjoydasju@gmail.com
* Correspondence: rajan.rajabalaya@ubd.edu.bn

**Abstract:** Curcumin is a hydrophobic compound with good anti-proliferative, anti-oxidative, and anti-cancer properties but has poor bioavailability. Liquid crystals (LC) can accommodate both hydrophilic and hydrophobic drugs. The aim of this study was to formulate and evaluate a novel vaginal drug delivery system for cervical cancer using a curcumin LC system. The curcumin LC system was formulated using surfactant, glycerol, and water together with curcumin. Three types of surfactants were used to optimize the formulation, i.e., Tween 80, Cremphor EL, and Labrasol. The optimized formulations were subjected to physicochemical analysis, and their efficacy was evaluated in HeLa cells. The pH of the formulations was in the range of 3.91–4.39. Environmental scanning electron microscopy (ESEM) observations revealed spherical as well as hexagonal micelles. In vitro release of LC curcumin from vaginal simulated fluid (VSF, pH 4.5) showed a release from 20.47% to 87.25%. The IC50 of curcumin in HeLa cells was 22.5 µg/mL, while the IC25 and IC75 were 6.5 µg/mL and 35µg/mL, respectively. The cytotoxicity of the formulations was determined in comparison with liquid crystals without curcumin and pure curcumin by performing a t-test based on a significance level of $p$ less than or equal to 0.05 ($p \leq 0.05$). The curcumin LC system was able to release the required amount of drug and was effective against the cervical cancer cell line examined.

**Keywords:** cervical cancer; liquid crystal; curcumin; HeLa cells

## 1. Introduction

Cervical cancer is characterized by abnormal cell growth in cervical tissues. Human papillomavirus (HPV), more prominently HPV16 and HPV18, is known to be the central causative agent of cervical cancer in 99% of cases [1–3]. Cervical cancer is the second leading cause of cancer death among women globally [4,5]. Cervical cancer is usually treated by means of surgery and radiotherapy followed by chemotherapy. In spite of having wide therapeutic applications, intravenous chemotherapy has many serious side effects [6]. These problems of systemic chemotherapy can be overcome by means of local or targeted specific chemotherapy, by administering drugs to cancer tissues while avoiding any systemic exposure and toxicity [7]. Recently, various targeted drug delivery systems have received attention for their ability of delivering drugs to cancer tissues in a highly selective manner without affecting other tissues. Such systems include nanoparticles [8], nanostructured lipid particles [9], liquid crystals (LC) [10], pH-responsive micelles [11], etc. LC have the ability to form an organized mesophase

exhibiting both crystalline solid and liquid properties [12]. The LC system has unique microstructure and physicochemical properties, whereby it can potentially solubilize oil and water-soluble compounds and thus can be used as a drug carrier system for both hydrophilic and hydrophobic drugs [13,14]. The incorporation of drugs in liquid crystals irrespective of their relative affinity towards hydrophilic or hydrophobic solvents is exploited in the field of biological sensing and drug delivery for various drug categories and formulations, i.e., analgesics, anti-cancer agents, drugs for liver diseases, anti-asthmatic drugs, nanoparticles formulations, and others [15]. LC allow sustained drug release and can be used to enhance drug permeation and accelerate drug absorption at the site of application [16].

Curcumin is a phenolic compound extracted from the rhizome of turmeric, *Curcuma longa*. Curcumin has been traditionally used to treat various allergic and inflammatory respiratory conditions and liver disorders and for wound healing [17]. In modern medicine, the antioxidative, anti-proliferative, and anti-inflammatory properties of curcumin have been explored and widely utilized to treat various medical conditions, such as inflammatory bowel syndrome, psoriasis, rheumatoid arthritis, and various cancers [18]. However, conventional preparations containing curcumin have limited therapeutic application due to the challenges imparted by some of the physicochemical properties of curcumin, especially its low aqueous solubility.

A work by Zhao et al. in 2004 found that curcumin inhibited, in a dose-dependent manner, cervical carcinoma growth studied in vitro using HeLa cells [4]. In another experiment conducted by the same group of researchers in 2007, HeLa cells were injected into mice and let grow for one week to form tumors [5] before curcumin treatment. In mice treated with curcumin, tumor growth was inhibited. The inhibition rate of curcumin was found to be 74.33% [4,5]. Rodero et al. reported that curcumin-loaded liquid crystalline systems with additional mucoadhesive properties are an efficient means of treating vulvovaginal candidiasis (VVC) upon vaginal administration [10]. Likewise, an in vitro study by Salmazi et al. (2015) showed better potency of a curcumin-loaded LC precursor mucoadhesive system consisting of chitosan and poloxamer 407 compared with a curcumin solution against VVC [19]. In 2017, Wang et al. constructed curcumin-encapsulated cubic liquid crystals using Brij 97 and IPM-PEG 400 to understand the phase and rheological properties of cubic LC [20].

So far, research studies have investigated the therapeutic properties against VVC of pure curcumin solutions, curcumin liquid crystals, or curcumin liquid crystals containing different combinations of Brij, chitosan, poloxamer, etc. Although a few research works have investigated the incorporation of natural anticancer agents in targeted carriers, according to our knowledge, a very low number of studies have been conducted on the use of curcumin-loaded liquid crystals for cancer treatment. Thus, the objective of the present research is to study liquid crystal systems for the effective delivery of curcumin. Different concentrations of surfactants such as Tween 80, Cremophor EL, Labrasol were tested to achieve a high safety profile and a gel-like consistency suitable for vaginal application. This is the first time these surfactants are used to produce liquid crystals and are tested for cervical carcinoma treatment. The curcumin-loaded LC system was prepared by the phase coacervation method, and the physicochemical evaluation of their organoleptic characteristics, pH, particle size, in vitro dissolution, and stability was carried out. Furthermore, the in vitro cytotoxicity of pure curcumin, unloaded LC system, and curcumin-loaded LC system was assessed in HeLa cell line.

## 2. Materials and Methods

Curcumin was purchased from Pi Chemicals Ltd. (Shanghai, China). The surfactant Tween 80 was acquired from EMD Millipore Corporation (Billerica, MA, USA). Glycerol, phosphate-buffered saline (PBS) tablets, and Dulbecco's modified Eagle's medium (DMEM) were obtained from Sigma-Aldrich, St Louis, MO, USA. Caprylocaproyl macrogol-8-glyceride (Labrasol®) was a kindly gift from Gattefossé (St Priest, France), and polyoxyl 35 castor oil (Cremophor® EL) from BASF (Ludwigshafen, Germany). All other chemicals and drug used were of analytical grade.

## 2.1. Curcumin-Loaded LC Preparation

An LC gel was prepared by the phase coacervation method by using different concentrations of surfactants such as Tween 80, Cremophor EL, and Labrasol (20%, 40%, 60%, or 80%) for the oil phase (A). The aqueous phase was composed of glycerin and water (B). Curcumin at different concentrations (0.5%, 1%, or 5%) was added and mixed well into the oil phase (A) at 40 ± 1 °C in a water bath with constant sonication. The aqueous phase (B) was heated up to 40 ± 1 °C and then was added to the oil phase. The formulations were cooled to room temperature and stored at 4 °C for further investigations. Formulations with different concentrations of surfactants were prepared to optimize the preparation. An ideal LC system for the purposes of this study should meet the following criteria: be a drug carrier system with a high safety profile and a gel-like consistency for vaginal application.

## 2.2. Physiochemical Evaluation of the LC System

### 2.2.1. Organoleptic Characteristic

Each formulation was tested for color and odor.

### 2.2.2. Determination of pH

A solution containing 0.5 g of the gel dissolved in 15 mL of neutralized distilled water (pH 7) was prepared, and the pH was measured using a pH meter (Mettler Toledo, OH, USA) [21].

### 2.2.3. Encapsulation Efficiency (EE)

The LC formulations were centrifuged at 12,000× *g* at a 5 °C for 1 h in two cycles using a cooling centrifuge (Eppendorf® Model 5810R, Eppendorf, Hamburg, Germany) to separate curcumin-containing vesicles from the unentrapped drug. The supernatant was then recovered and lysed with methanol. The sediment was filtered through a 0.45 μm nylon disk filter. The free drug concentration in the supernatant was determined using reverse-phase HPLC (1200 HPLC series; Agilent Technologies, Santa Clara, CA, USA) with a C-18 analytical column (5 μm, 4.6 × 250 mm; Phenomenex, Torrance, CA, USA). The mobile phase was a 50:50 (*v/v*) mixture of acetonitrile and 2% acetic acid in water at a flow rate of 1.2 mL min$^{-1}$. The working solution of curcumin was analyzed by HPLC using a C18 column with UV detection at 425 nm [22].

$$\% \text{ Encapsulation efficienc} = \frac{\text{Total drug content} - \text{drug content in supernatant}}{\text{Total drug content}} \times 100 \qquad (1)$$

### 2.2.4. Particle Size Determination

The LC preparations (0.1 g) were diluted in 5 mL of double-distilled water and sonicated for 30 s in an ice bath. Analysis of the samples was carried out at a scattering angle of 173° and a temperature of 250 °C [21]. Then, 1 mL of each solution was injected into a Zetasizer cuvette and analyzed in a Malvern Zetasizer Nano ZS (Malvern Instruments Limited, Worcestershire, United Kingdom).

## 2.3. FTIR and ATR–FTIR Analysis

The IR spectra of the drug were recorded using FTIR spectrometer (Shimadzu FTIR-8400S, Kyoto, Japan). The samples were prepared as compressed KBr disks [23]. The IR analysis was performed in the frequency range of 500–4000 cm$^{-1}$. The formulations were tested for drug content using the ATR–FTIR method using a Shimadzu IR Prestige 21 equipment. The samples were placed on the sample holder and run over the frequency range of 500–4000 cm$^{-1}$.

## 2.4. Environmental Scanning Electron Microscopy (ESEM)

A small amount of a selected LC gel was placed on a holding disk, and a drop of distilled water was added. The disk was placed in a chamber inside an environmental scanning electron microscope (ESEM, Quanta 450 FEG; by Fei Company, Hillsboro, OR, USA). Water within the sample was allowed to evaporate before the examination. The sample was placed in the sample holder and scanned [14].

## 2.5. Dissolution Study

The in vitro dissolution study of different LC gels was carried out by using a dialysis bag consisting of a cellophane membrane. Vaginal simulated fluid (VSF, pH 4.5) was used for the study to simulate normal adult vaginal conditions. An amount of each LC gel formulation (0.5 g) was placed in VSF maintained at 37 ± 0.5 °C with constant stirring at 50 rpm [24]. Samples were taken every hour for eight hours. The samples were diluted with buffer, and the absorbance was determined using HPLC (1200 HPLC series, Agilent Technologies, Santa Clara, CA, USA) using the method previously described.

## 2.6. Stability Studies

The formulations were stored in universal bottles at three different temperatures, i.e., 4 °C, 25 ± 2 °C, and 37 °C. The drug content was estimated every 15 days for 45 days to identify any changes in the formulation [25]. The formulations were kept in vials sealed with aluminum foil. After three months, the EE was determined as described above [21,22].

## 2.7. Cytotoxicity of Curcumin, Liquid Crystal System, and Curcumin-Loaded Liquid Crystal System in HeLa Cells

### 2.7.1. MTT Assay

Cervical cancer HeLa cells were trypsinized, resuspended in complete medium, and seeded in 96-well plates at a density of $1 \times 10^4$ cells/well. The cells were then exposed to various concentrations of curcumin. At the end of the treatment, 10 μL of MTT solution (5 mg/mL) was added into each well, and incubation was carried out for 4 h at 37 °C. After that, the culture medium was removed and replaced with DMSO to dissolve the purple formazan crystals that had formed. Finally, the absorbance of the solution was measured spectrophotometrically at 570/630 nm with a microplate reader.

### 2.7.2. Determination of Half-Maximal Inhibitory Concentrations (IC50) of Curcumin in HeLa Cells

HeLa cells were incubated in a 96-wells microtiter plate before being treated with curcumin at different concentrations in DMEM (0.78125 μg/mL, 1.5625 μg/mL, 3.125 μg/mL, 6.25 μg/mL, 12.5 μg/mL, 25 μg/mL, 50 μg/mL, and 100 μg/mL). The IC50 of the compound was determined with the MTT assay. The IC25 and IC75 were also determined [26,27].

### 2.7.3. Determination of the Safety Profile of the LC Preparations

The study was done for liquid crystals that were prepared with different surfactants. Liquid crystals prepared using Tween 80, Cremophor EL and Labrasol without curcumin were diluted in DMEM at different concentrations and used to treat HeLa cells in 96-wells microtiter plates. The safety profile of the liquid crystal was determined with the MTT assay [28].

### 2.7.4. Determination of the Efficacy of the Curcumin Liquid Crystal Preparations

The curcumin liquid crystal preparations were diluted in DMEM at three concentrations according to IC25, IC50, and IC75 of curcumin and then used to treat HeLa cells [29].

*2.8. Statistical Analysis*

The statistical analysis was conducted using SPSS v8.0 (IBM, Armonk, NY, USA). All data are reported as means ± standard deviation for experiments regarding the drug efficacy study. Statistical difference between control and treated groups was determined using a simple *t*-test. A *p*-value less than or equal to 0.05 ($p \leq 0.05$) was considered as statistically significant [30].

## 3. Results

*3.1. Selection of a Suitable Surfactant for the Liquid Crystal System*

The organoleptic characterization and treatment of LC were conducted in HeLa cells using LC prepared with the surfactants Tween 80, Cremophor EL, and Labrasol at different concentrations. It was found that Cremophor EL and Labrasol at different concentrations were highly cytotoxic as assessed with the MTT assay in HeLa cells; thus, they were eliminated from the study. Tween 80 was selected as the ideal surfactant, as it also has better organoleptic properties than the other surfactants.

*3.2. Physiochemical evaluation of the LC system*

Organoleptic properties, pH, encapsulation efficiency, and particle size of the liquid crystal formulations were evaluated.

The organoleptic characteristics of the liquid crystal formulations are presented in the Table 1. As mentioned above, the chosen formulations showed a brown to yellowish brown color with an oily odor. The pH of the formulations was in the range of 3.91–4.39 and thus appropriate for vaginal preparations. The encapsulation efficiency ranged from 80.36 to 93.26. The mean particle size ranged from 6.12 to 11.01 nm.

**Table 1.** Organoleptic properties, particle size, and encapsulation efficiency of the liquid crystal formulations.

| Formulation Code | Curcumin: Tween 80 (%) | Colour | Odour | pH | Encapsulation Efficiency (%) ± SD | Vesicle Size (nm) ± SD | Polydispersity (PDI) Index | Zeta Potential (mV) |
|---|---|---|---|---|---|---|---|---|
| A1 | 0.5:40 | Brown | Oil-like | 3.91 | 91.02 ± 1.69 | 8.01 ± 0.85 | 0.158 ± 0.07 | −28.2 ± 1.42 |
| A2 | 1.0:40 | Brown | Oil-like | 4.13 | 92.31 ± 2.36 | 10.31± 1.81 | 0.102 ± 0.04 | −23.4 ± 2.27 |
| A3 | 5.0:40 | Yellowish brown | Oil-like | 4.24 | 81.58 ± 1.57 | 11.01 ± 1.30 | 0.441 ± 0.12 | −18.1 ± 1.47 |
| B1 | 0.5:60 | Brown | Oil-like | 4.18 | 88.01 ± 2.80 | 6.12 ± 0.67 | 0.461 ± 0.17 | −19.5 ± 1.89 |
| B2 | 1.0:60 | Brown | Oil-like | 4.07 | 93.26 ± 3.05 | 7.56 ± 1.21 | 0.254 ± 0.09 | −22.5 ± 1.43 |
| B3 | 5.0:60 | Yellowish brown | Oil-like | 4.39 | 80.36 ± 2.46 | 12.97 ± 1.12 | 0.458 ± 0.13 | −16.8 ± 1.22 |

*3.3. FTIR/ATR–FTIR Analysis*

ATR–FTIR was performed on the samples B1 and B2, which had the highest encapsulation efficiency % and the lowest vesicle size amongst the B series samples prepared with 60% Tween 80. The samples' spectral images are shown in Figures 1 and 2. The characteristic peaks of curcumin are visible at 3448, 3120–3520, 1427, 1512, 1604, 1627, and 1762 cm$^{-1}$. Other observed absorbance peaks were: aromatic –OH stretching, located in the range of 3200–3500 cm$^{-1}$, C=O stretching at 1627cm$^{-1}$, C=C stretching at 1512 cm$^{-1}$, C=C aromatic stretching at 1427 cm$^{-1}$, phenol –C–O– stretching at 1280.78 zcm$^{-1}$, and phenyl –CH– bending (in plane) at 1153.49 cm$^{-1}$. The peak in the range 2200–2300 cm$^{-1}$ was due to carbon dioxide which was incorporated during preparation of the KBr disk due to a technical error.

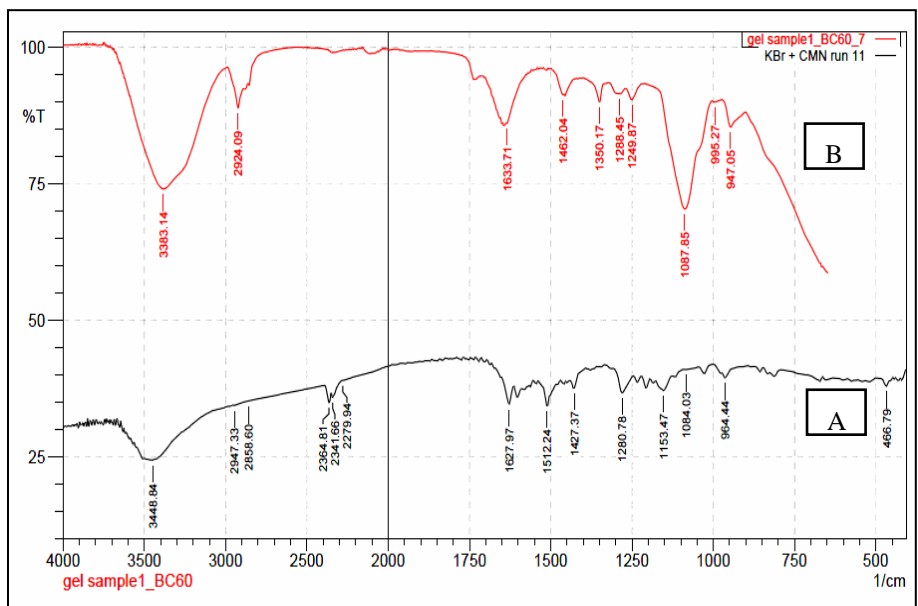

**Figure 1.** ATR–FTIR spectra of pure curcumin (**A**) and of the B1 curcumin gel (**B**).

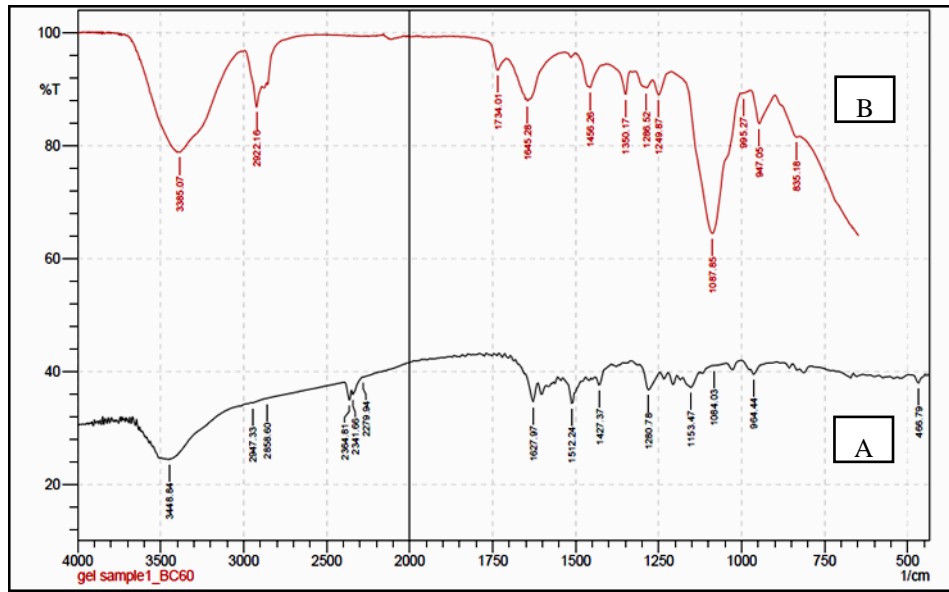

**Figure 2.** ATR–FTIR spectra of pure curcumin (**A**) and the B2 curcumin gel (**B**)

### 3.4. ESEM

The structural shapes of the A1, A2, B1, and B2 formulations were determined by environmental scanning electron microscopy. The images are presented in Figure 3A–D and show a spherical structure of the liquid crystalline gel formulations. It was also observed that there were spherical as well as hexagonal micelles which were distributed uniformly throughout the formulation.

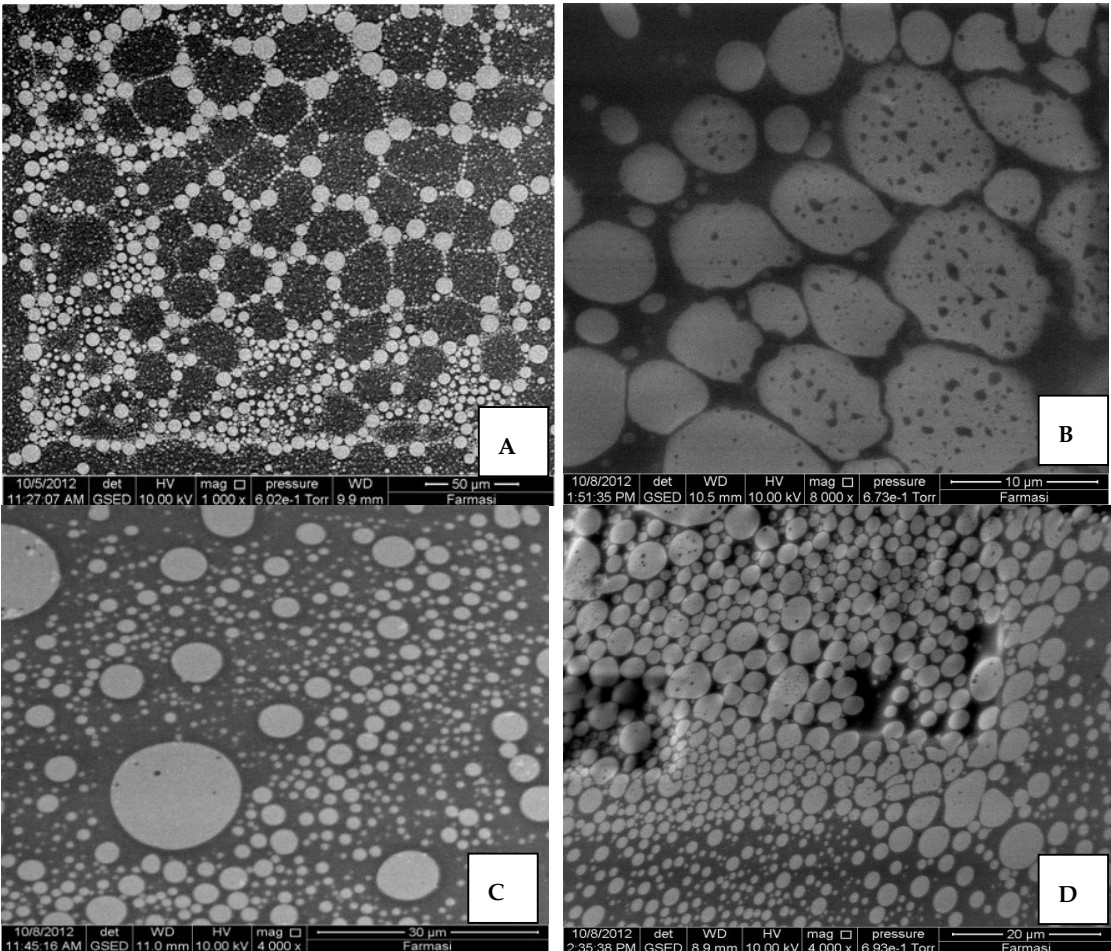

**Figure 3.** Environmental scanning electron microscopy (ESEM) of (**A**) formulation A1, (**B**) A2 formulation, (**C**) B1 formulation, and (**D**) B2 formulation.

*3.5. Dissolution Study*

Drug release vs. time was plotted for all formulations, as presented in Figure 4. The release of the curcumin LC gel from all formulations was controlled for 8 h. In vitro release of LC curcumin from the VSF, pH 4.5, was between 20.47% and 87.25%. The drug release kinetics of the formulations are given in Table 2.

**Table 2.** Drug Release kinetics for check point formulations (CPFs) and for optimized batch (BO).

| Formulation Code | Zero Order | | First Order | | Higuchi Model | | Korsmeyer–Peppas | | |
|---|---|---|---|---|---|---|---|---|---|
| | $K_o$ $(h^{-1})$ | $r^2$ | $K_1$ $(h^{-1})$ | $r^2$ | $K_H$ $(h^{-1/2})$ | $r^2$ | $K_{KP}$ $(h^{-n})$ | $r^2$ | $n$ |
| A1 | 4.6318 | 0.9734 | −0.0335 | 0.9897 | 30.7499 | 0.9448 | 0.2087 | 0.9950 | 0.46 |
| A2 | 5.6309 | 0.9275 | −0.0540 | 0.9734 | 28.3905 | 0.9323 | 0.3088 | 0.9902 | 0.42 |
| B1 | 5.7982 | 0.9923 | −0.0545 | 0.9867 | 32.7567 | 0.9528 | 0.2913 | 0.9793 | 0.41 |
| B2 | 7.4150 | 0.9943 | −0.1002 | 0.9485 | 32.6238 | 0.9547 | 0.3379 | 0.9819 | 0.44 |

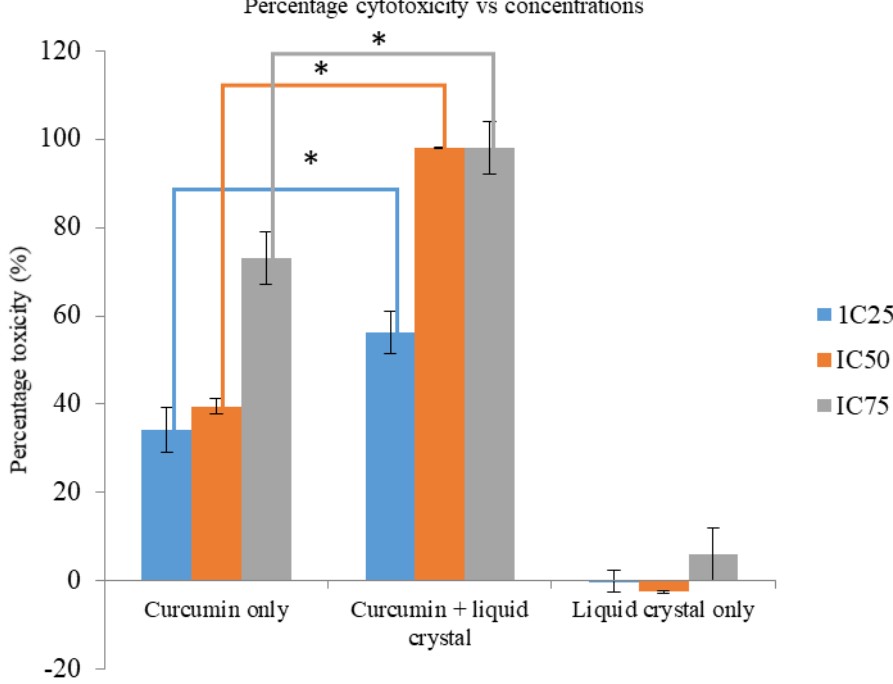

**Figure 4.** Drug release profile for all formulations determined in PBS.

### 3.6. Stability Studies

Stability studies were carried out by storing four formulations, A1, A2, B1, and B2, of curcumin LC gels at various temperatures, i.e., 5 ± 2 °C, 25 ± 0.5 °C and 45 ± 0.5 °C. The drug content was estimated before storing [25]. The formulations were kept in vials sealed with aluminum foil. After three months, the EE was determined as described above [21,22]. The drug contents of the samples are presented in Table 3.

**Table 3.** Drug content determined in stability studies.

| Formulation Code | Period | Drug Content (%) | | |
| --- | --- | --- | --- | --- |
| | | 5 ± 2 °C | 25 ± 0.5 °C | 45 ± 0.5 °C |
| A1 | Initial | 91.02 ± 1.69 | 91.02 ± 1.69 | 91.02 ± 1.69 |
| | After 3 months | 88.01 ± 1.11 | 87.57 ± 1.87 | 85.02 ± 2.35 |
| A2 | Initial | 92.31 ± 2.36 | 92.31 ± 2.36 | 92.31 ± 2.36 |
| | After 3 months | 89.41 ± 2.06 | 87.23 ± 1.36 | 85.87 ± 1.57 |
| B1 | Initial | 88.01 ± 2.80 | 88.01 ± 2.80 | 88.01 ± 2.80 |
| | After 3 months | 86.98 ± 1.40 | 84.08 ± 2.94 | 82.14 ± 13.57 |
| B2 | Initial | 93.26 ± 3.05 | 93.26 ± 3.05 | 93.26 ± 3.05 |
| | After 3 months | 91.23 ± 2.10 | 89.02 ±2.23 | 85.52 ± 3.67 |

### 3.7. Cytotoxicity of Curcumin, Liquid Crystal System, and Curcumin-Loaded Liquid Crystal System

Six formulations were prepared with Tween 80, out of which four were tested, namely, A1, A2, B1, and B2. The B2 formulation had the highest encapsulation efficiency and best drug release profile amongst all formulations. Also, it had acceptable levels of vesicle size, polydispersity index, zeta potential, and stability. Thus, B2 was chosen for further investigations in inhibitory concentration studies.

The determination of the IC50 of curcumin in HeLa cells was conducted with triplicate samples. The IC50 of curcumin in HeLa cells was found to be 22.5 μg/mL, while the IC25 and IC75 were 6.5 μg/mL and 35 μg/mL, respectively. From Figure 5, it is observed that treatment of HeLa cells with

the curcumin liquid crystal system showed significant efficacy when compared to treatment with pure curcumin. As for the blank liquid crystal system, the cytotoxicity was negligible.

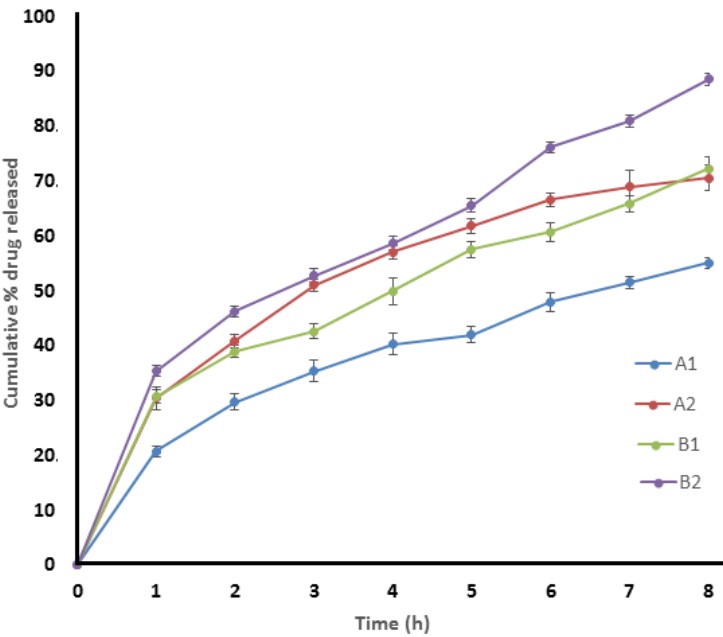

**Figure 5.** Cytotoxicity of curcumin, curcumin-loaded liquid crystal system, and liquid crystal (*n* = 3). * *p* < 0.05

## 4. Discussion

### 4.1. Organoleptic Properties of Liquid Crystal Formulations

LC prepared with Cremophor EL showed cytotoxicity at high concentration. This finding is supported by the results of Zhang et al. who reported that Cremophor EL was toxic and caused hypersensitivity reactions in certain individuals [5]. Although LC prepared with Labrasol showed an average safety profile, they were not considered in this study, as they did not form a preparation of the desired consistency and encapsulation efficiency. Liquid crystals prepared with 40% and 60% of Tween 80 were chosen in this study for subsequent analyses, because of their convenient physical properties. Many studies have reported that Tween 80 is suitable to produce nanoparticles, nanoemulsions, and self-microemulsifying drug delivery systems [31,32].

### 4.2. pH Measurement

The LC formulations contained a very limited quantity of aqueous phase; hence, their pH was determined for quality control and to analyze their compatibility with vaginal pH. All the Tween 80 formulations were compatible and were within the range of normal vaginal pH [33].

### 4.3. Encapsulation Efficiency

Encapsulation efficiency is a critical factor that influences the stability of LC. The concentrations of surfactants affect the morphology, size, and drug loading capacities of the vesicles. Inappropriate concentrations may vary the shape of the vesicles leading to drug leakage before the vesicles reach the target tissues [34]. EE also depends on the amount of oil with respect to those of surfactant and water. The encapsulation of curcumin is mainly due to its dissolution in the molten oil phase and its partition between the aqueous phase and the surfactant. The high EE capacities of all formulations may be due to a low bilayer permeability and high lipophilic bilayer thickness. This may be helpful to achieve high drug loading in the core of the lipophilic bilayer [35]. In general, the EE was reduced

when raising the concentration of surfactant. High amounts of Tween 80 may increase the viscosity of the hydrophilic phase, which leads to controlled drug release from the vesicles [22]. In fact, Tween 80 is more hydrophilic than lipophilic, which favors its solubility in the aqueous phase thus increasing the viscosity of LC vesicles; this promotes an even distribution of the drug within hydrophobic chains that form the bilayer structure of the LC [21,35].

### 4.4. Particle Size, zeta Potential, and Polydispersity Index

The size and distribution of the vesicles of the LC formulations are of key importance for targeted drug delivery and a suitable permeation profile [34]. A PDI value of less than 0.1 denotes a homogenous particle population, while a PDI greater than 0.3 indicates increased heterogeneity [36]. In this study, three formulations were shown to have PDI values higher than 0.3. The vesicle size ranges were heterogeneous and within the required limits.

The zeta potential values of Tween 80-based LC formulations were high, probably due to the low molecular weight of Tween 80. It is evident in this study that the upper negative zeta value of Tween 80-based formulations prevented particle–particle interactions [37,38], whereas formulations with low amounts of Tween 80 and low zeta potential values promoted particle aggregation. This is demonstrated by the lesser particle size of Tween 80 than higher ratio formulations.

### 4.5. FTIR

The spectra of the surfactant, curcumin, and Tween 80/curcumin formulations demonstrated that there were no significant peak shifts with respect to the spectra of the pure components of the formulations. This shows that the chemical functional groups of the drug and the surfactants were intact after mixing the components to obtain the LC formulations. The spectra showed few characteristic peaks of the drug overlapping with those of the surfactant, which may be due to the encapsulation of the drug within the surfactant's bilayer.

### 4.6. ESEM

ESEM demonstrated that the LC formulations with a low concentration ratio of curcumin and Tween 80 were characterized by a smaller spherical shape compared to formulations with higher ratios of curcumin and Tween 80. In formulations with a low concentration of Tween 80, the shape of the vesicles was not as spherical as that of vesicles in formulations with higher concentrations of Tween 80, which may be due to the smaller size of the particles. Repulsion between the surfactant head groups could have improved the curvature of the micelles, establishing spherical and hexagonal vesicles [39]. This is supported by the higher values of the zeta potential in the range between −16 and −28 mV, indicating a good dispersion of the LC vesicles in Tween 80-based formulations diluted with distilled water [40]. Spherical formations of micelles were observed in all formulations. However, it is not clear whether the micelles formed were normal or reversed-phase micelles. A more powerful microscope such as a cryo-transmission electron microscope (cryo-TEM) is required in order to observe the micelles phase and structure in more detail.

### 4.7. Dissolution Study

The in vitro dissolution profiles of the formulations A1, A2, B1, B2 were examined in PBS and shown in Figure 5. It is clear that in the case of formulation A1 (drug/Tween 80 ratio = 0.5:40), the drug release was approximately 54%, and in the case of formulation A2 (drug/Tween 80 ratio = 1.0:40), the drug release was approximately 70% after 8h. The cumulative percentage of drug release after 8 h was higher for the formulation A2 because of a higher drug encapsulation efficiency compared to the formulation A2. Similarly, in the case of formulation B2 (drug/Tween 80 ratio = 1.0:60), the cumulative percentage of drug release was 88.35%, higher than that of formulation B1 (drug/Tween 80 ratio = 0.5:60) after 8 h. For in vitro dissolution, the A3 and B3 formulations were not analyzed due to their low drug EE. A dissolution study was carried out in VSF and it was observed that the drug release

pattern was similar to that determined in PBS. It was also shown that the smaller vesicle size of the formulations, the higher the drug release.

### 4.8. Kinetic Analysis of In Vitro Dissolution Study

The drug release data were analyzed to study the drug release mechanism. A curve-fitting analysis was done for the four formulations. The release data were fitted to various mathematical models, i.e., zero-order, first-order, Higuchi, and Korsmeyer–Peppas models, to examine the path and the mechanism of drug release. It was observed that the release of drug from the different formulations occurred through different mechanisms. The drug release pattern from A1 fitted better with the Korsmeyer–Peppas ($r^2$, 0.9950), the first-order ($r^2$, 0.9897), and the zero-order ($r^2$, 0.9734) equations than with the Higuchi equation. The drug release pattern from A2 was best explained by Korsmeyer–Peppas and first-order equations. The drug release patterns from B1 and B2 was best explained by zero-order equations.

### 4.9. Stability Studies

Upon physical inspection of each formulation on day 0, 15, 30, 60, and 90, it was observed that the formulations were most stable when stored at 5 °C. There were signs of drug segregation in formulation stored at 25 ± 0.5 °C and 45 ± 0.5 °C, especially in formulations with 5% curcumin. Drug segregation was seen as early as day 15 and continued to progress on subsequent inspections. At higher temperature, the liquid crystals were less stable, and the rate of drug degradation was higher. The formulations stored at 25 and 45 °C showed reduced drug content, whereas the formulations stored at 5 °C did not show any decrease in drug content. The stability results were highly significant, with $p < 0.03$.

### 4.10. Determination of IC50 of Curcumin in HeLa Cells

In previous studies, it was found that the bioavailability of curcumin after an oral dose was very low even after the administration of a high dose of curcumin daily (8 g/day) [18]. However, Zhao et al. proved that curcumin can be used in vivo to inhibit cervical cancer cell growth in a mouse model [4]. Taking this information into account, it would be useful to prepare a curcumin formulation for other routes of administration. By incorporating curcumin into an LC system, the resulting preparation can be applied topically, achieving a sustained release of drug to treat cervical cancer. A site-specific application is preferable in the treatment of cervical cancer compared to a systemic treatment. In fact, a site-specific application can ensure the rapid distribution of the drug inside the vaginal area, while avoiding first-pass effects that usually occur when taking a medication orally [41]. It was observed that treatment of HeLa cells with a curcumin liquid crystal system was significantly more effective compared to the treatment with pure curcumin. Also, the cytotoxicity of the unloaded liquid crystal system was negligible.

## 5. Conclusions

Curcumin liquid crystal formulations have appropriate pH and suitable physicochemical properties for vaginal application. These formulations dissolve conveniently, in relation to their morphological structure. The curcumin liquid crystal system releases the required amount of drug and is effective in inhibiting the growth of a cervical cancer cell line. Thus, curcumin liquid crystal system formulations should be evaluated in vivo for their anti-tumor properties in the treatment of cervical cancer.

**Author Contributions:** Conceptualization, S.R.D.; and R.R.; methodology, S.R.D.; software, C.-W.M.; validation, S.R.D.; S.K.D.; and R.R.; formal analysis, S.K.D; investigation, N.A.B.A.; and K.R.Y.; resources, S.R.D.; data curation, N.A.B.A.; writing—original draft preparation, N.A.B.A.; and R.R.; writing—review and editing, S.R.D.; visualization, N.A.B.A.; supervision, S.R.D.; project administration, R.R.; funding acquisition, S.R.D.; All authors have read and agreed to the published version of the manuscript.

**Funding:** The authors would like to acknowledge International Medical University for the funding and laboratory facilities. (IMUPharm 2013-11).

**Acknowledgments:** The authors would like to acknowledge International medical University for the project and laboratory facilities.

**Conflicts of Interest:** All authors declare that they have no conflict of interest.

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
