# Peer review of "Development and Evaluation of Curcumin Liquid Crystal Systems for Cervical Cancer"

_scipharm, doi:10.3390/scipharm88010015_

Round 1

Reviewer 1 Report

The manuscript presents an interesting study regarding the development of a new formulation of curcumin for the treatment of cervical cancer. The manuscript has some points that should be clarified before publication.

The introduction section can be updated with the last researches regarding cervical cancer etiology, see DOI: 10.3892/ijo.2018.4256

In the phrase: “[18] reported that curcumin loaded liquid crystalline systems 61 with additional mucoadhesive property can solve such kind of problems and is an efficient means of 62 treating vulvovaginal carcinoma (VVC) upon vaginal administration [18]” please modify with “Rodero et al. reported…..”

In the material and methods describe briefly the MMT assay .

The part “Selection of suitable surfactant for f liquid crystal system”, “The percentage cytotoxity (%) of different surfactant such as tween 80, cremophor EL and 166 labrasol at different concentrations. The organoleptic characterization and treatment on HeLa cells 167 were conducted on LC prepared with above surfactants. The tween 80 was selected as the ideal 168 surfactant compared other surfactant.” Is not clear.  In this section you have to present clear the results based on which you chose the composition.

In line 180 you state ” ATR- FTIR test were performed with B1 and B2 samples”, why not also for all the other samples from the table 1?

Line 192 you state: “The structural shapes are taken for the A1, A2, B1 and B2 formulations photograph”. What is the criteria for evaluating only 3 of the 6 formulations?

In Table 2 you have the drug release kinetics only for 4 formulations. It is not clear how these formulations were chosen from the 6 ones.

In line 207 you state that the stability study was carried only on 2 formulations: B1 and B3 and in table 3 you present the data of 4 formulations: A1, A2, B1, B2 but B3 is absent.  Please clarify

Please clarify which formulation you tested for cytotoxicity on HeLa cells.  

Reviewer 2 Report

The article presented by the authors fits to Sci. Pharm. scope. The article is written concise and systematized, and the results are well presented.

However, some similar articles can be found in the literature. For example, Int J Nanomedicine. 2015; 10: 4815–4824 by Salmazi et al. describe a similar system with the same active substance. Also, other works (Suhaimi, H., Ahmad, F. B., & Friberg, S. E. (1995). Curcumin in a model skin lotion formulation. Journal of pharmaceutical sciences, 84(3), 376-380.; Wang, Z., Fu, L., Liu, X., Zhang, L., Guo, F., & Zhao, X. (2017). Phase and Rheological Properties of a Curcumin-Encapsulated Cubic Liquid Crystal. Journal of Surfactants and Detergents, 20(3), 673-679.) can be found already published on the topic in the literature.  Thus, the authors are  asked to discuss the literature and add some  clarifications regarding the choice and the differences between the formulations. Also, the novelty and originality of the article should be discussed and clarified.

Please find below other commentaries:

rows 16-17 - please correct the phrase

2.5 - the assay method should be presented!

Methods from section 2.7 should be briefly presented.

2.8. Statistical analysis (row159) - please indicate the software.

rows 164 - 168 - please present in more detail. The data can be added as supplementary material.

Section 3.6. and 2.8  please indicate the number of experiments for each determination. This data should be indicated in the text and in the descriptions of figures and tables.

Section 4.5. FTIR analysis is useful for the characteriazation of the formulation. The fingerprint is useful only for the identification of the product. For example it should a parameter analyzed in the stability data.  The method can't demonstrate the compatibility or incompatibility of the components in the mixture. Please change the paragraph accordingly. 

Round 2

Reviewer 1 Report

The authors assessed all the comments of the reviewers and now the manuscript is much more improved compared with the first version. 

Reviewer 2 Report

The manuscript has been substantially improved and can be published in Sci. Pharm.